# Exploiting CRISPR-Cas to manipulate *Enterococcus faecalis* populations

**Karthik Hullahalli[†], Marinelle Rodrigues[†], Kelli L Palmer\***

Department of Biological Sciences, The University of Texas at Dallas, Richardson, United States

**Abstract** CRISPR-Cas provides a barrier to horizontal gene transfer in prokaryotes. It was previously observed that functional CRISPR-Cas systems are absent from multidrug-resistant (MDR) *Enterococcus faecalis*, which only possess an orphan CRISPR locus, termed CRISPR2, lacking *cas* genes. Here, we investigate how the interplay between CRISPR-Cas genome defense and antibiotic selection for mobile genetic elements shapes in vitro *E. faecalis* populations. We demonstrate that CRISPR2 can be reactivated for genome defense in MDR strains. Interestingly, we observe that *E. faecalis* transiently maintains CRISPR targets despite active CRISPR-Cas systems. Subsequently, if selection for the CRISPR target is present, toxic CRISPR spacers are lost over time, while in the absence of selection, CRISPR targets are lost over time. We find that forced maintenance of CRISPR targets induces a fitness cost that can be exploited to alter heterogeneous *E. faecalis* populations.

## Introduction

*Enterococcus faecalis* is a Gram-positive opportunistic pathogen that commensally inhabits the gastrointestinal tracts of humans and other mammals (*Lebreton et al., 2014*). Enterococcal infections are considered serious public health threats, and rising antibiotic resistance makes these infections particularly difficult to treat (*Arias and Murray, 2012*; *Kristich et al., 2014*; *Centers for Disease Control and Prevention, 2014*; *Sievert et al., 2013*). Sequence analysis of multidrug-resistant (MDR) isolates of *E. faecalis* reveals that they typically possess expanded genomes relative to more drug-sensitive isolates and have acquired large segments of mobile DNA in the form of prophage, genomic islands, transposons, and plasmids (*Raven et al., 2016*; *Palmer et al., 2012*; *Paulsen et al., 2003*; *Bourgogne et al., 2008*). These extraneous DNA elements often encode antibiotic resistance determinants and virulence factors that facilitate host infection and colonization, thus making the horizontal dissemination of DNA one of the prime causative factors for the emergence of MDR *E. faecalis* (*Palmer et al., 2014*).

Clustered Regularly-Interspaced Short Palindromic Repeats and their associated Cas proteins (CRISPR-Cas) are adaptive immune systems employed by bacteria to reduce the prevalence of mobile genetic elements (MGEs), such as bacteriophage and plasmids, in their populations (*Rath et al., 2015*). Some *E. faecalis* encode Type II CRISPR-Cas systems, defined by the presence of *cas9*, and which consist of two main components: a CRISPR array and *cas* genes. The mechanism of Type II CRISPR-Cas systems has been well characterized (*Marraffini, 2015*; *Sapranauskas et al., 2011*; *Nishimasu et al., 2014*; *Jinek et al., 2014*; *Anders et al., 2014*; *Deltcheva et al., 2011*). The CRISPR array consists of 36 bp repetitive DNA elements (repeats) interspersed by 30 bp sequences usually derived from foreign DNA (spacers). The cognate spacer sequence present in foreign DNA, termed the protospacer, is usually located proximally to a conserved DNA sequence referred to as the protospacer adjacent motif (PAM). The final spacer in the CRISPR array (terminal spacer) is followed by a degenerated repeat (terminal repeat). During CRISPR interference, the Cas9

**\*For correspondence:** kelli.palmer@utdallas.edu

[†]These authors contributed equally to this work

**Competing interests:** The authors declare that no competing interests exist.

endonuclease is guided to DNA targets by CRISPR RNAs (crRNAs), which are processed transcripts derived from the CRISPR array. An active Cas9-crRNA targeting complex is also associated with a trans-activating crRNA (tracrRNA), which is partially complementary to the repeats of the CRISPR array. This targeting complex samples PAMs in DNA, and once it encounters a match to the affiliated crRNA spacer sequence, Cas9 creates a double-stranded break in the target DNA. Three Type II CRISPR occur with variable distributions in the *E. faecalis* species, including two that possess *cas* genes (CRISPR1 and CRISPR3, which are Type II-A systems [*Fonfara et al., 2014*]), and one orphan array (CRISPR2) that is ubiquitous but lacks associated *cas* genes (*Bourgogne et al., 2008*; *Palmer and Gilmore, 2010*; *Hullahalli et al., 2015*). The orphan locus possesses repeats identical to those in CRISPR1, but not CRISPR3. Further, the consensus PAM sequences for CRISPR1-Cas and CRISPR are identical (NGG), while the PAM for CRISPR3 is distinct (NNRTA) (*Price et al., 2016*). The *cas9* associated with CRISPR3 is distinct in sequence and function from the CRISPR1 *cas9*. We have previously shown that CRISPR3-Cas9 cannot confer defense from conjugative plasmids using CRISPR2 spacers (*Price et al., 2016*).

An inverse correlation between the occurrence of Type II CRISPR-Cas systems and antibiotic resistance has been reported for *E. faecalis* (*Palmer and Gilmore, 2010*). All *E. faecalis* isolates possess a CRISPR2, but most MDR strains lack the functional CRISPR1-Cas or CRISPR3-Cas systems (*Palmer and Gilmore, 2010*; *Hullahalli et al., 2015*). In this study, using conjugation assays, we find that CRISPR2 is functional for sequence interference in otherwise *cas*-deficient MDR *E. faecalis* isolates upon the introduction of CRISPR1-*cas9*. We provide a mechanism for the ubiquitous presence of CRISPR2 in MDR enterococci despite the locus being natively inactive for genome defense in those strains. Intriguingly, we also find that *E. faecalis* can temporarily tolerate CRISPR-targeted plasmids, and that strains forced to maintain CRISPR targets exhibit a growth defect and can be outcompeted by other strains. We present a novel approach to alter the structure of *E. faecalis* populations by exploiting CRISPR-Cas, selection, and intraspecies competition. This approach could be applied in the future to reduce intestinal carriage of MDR *E. faecalis* in high-risk patients (*Al-Nassir et al., 2008*; *Ubeda et al., 2010*; *Clutter et al., 2013*; *Rice, 2013*).

## Results

### Design of a rapidly modifiable conjugation assay to study CRISPR function in *E. faecalis*

V583 is a model MDR *E. faecalis* strain, and was the first fully sequenced vancomycin-resistant isolate in the U.S. (*Paulsen et al., 2003*). It possesses an expanded genome relative to commensal isolates, and its sole CRISPR locus is the orphan CRISPR2 (*Palmer et al., 2012*; *Bourgogne et al., 2008*). We have previously reported that CRISPR-Cas lowers the frequency of acquisition of large (50–70 kbp) conjugative pheromone responsive plasmids (PRPs) in *E. faecalis* T11, a strain closely related to V583 but lacking the HGT-driven genome expansion characteristic of V583 (*Price et al., 2016*). However, it remained to be determined whether the CRISPR2 locus of MDR strains possessing different spacer dispositions was also active. In order to test the functionality of CRISPR-Cas across a wide range of isolates, we developed a method to rapidly create protospacer-bearing conjugative plasmids. First, we inserted the *oriT* sequence from the pheromone-responsive plasmid pCF10 into the cloning vector pLZ12 (*Perez-Casal et al., 1991*), creating pKH12. pKH12 was linearized via PCR using primers with overhangs engineered to introduce different protospacers with the CRISPR1/CRISPR2 NGG PAM sequence (*Price et al., 2016*) to generate the pKHSX series, where X defines a spacer from our CRISPR dictionary (*Hullahalli et al., 2015*). We then examined a series of *E. faecalis* strains to identify suitable plasmid donors. CK111SSp(pCF10-101) (*Kristich et al., 2007*) was found to be an effective donor that was able to mobilize pKH-derivatives into a variety of strains. Since CK111SSp(pCF10-101) encodes a CRISPR1-Cas system, we also generated strain C173, with *ermB* disrupting the CK111SSp(pCF10-101) *cas9* locus. C173 was used as a donor to analyze the functions of spacers that are also present in the CK111SSp(pCF10-101) CRISPR1 or CRISPR2 arrays. The experimental scheme for conjugation assays is shown in *Figure 1—figure supplement 1*.

When possible, conjugation frequencies were determined as the ratio of transconjugants per donor, as is routine for the field. However, *E. faecalis* Merz96 and DS5 both showed bactericidal activity against CK111SSp(pCF10-101). Zones of inhibition were observed after spotting overnight

cultures of Merz96 and DS5 onto CK111SSp(pCF10-101) lawns (data not shown). For these strains, conjugation frequencies were determined as the ratio of transconjugants per recipient.

## CRISPR1-Cas and CRISPR2 are functionally linked across diverse *E. faecalis* lineages

CRISPR1-*cas* and CRISPR2 are the most prevalent CRISPR loci in *E. faecalis* (*Hullahalli et al., 2015*). These CRISPR arrays contain identical repeats and differ only in their conserved locations in the chromosome and in the sequence of the terminal repeat (discussed later). We sought to confirm whether we could detect CRISPR1-Cas function using conjugation assays with pKHSX plasmids and if this assay could be used across genetically diverse *E. faecalis* isolates. We first assessed ATCC 4200RF, isolated from the bloodstream of a rheumatic fever patient in 1926, two years prior to the discovery of penicillin (*Birkhaug and Schilling, 1927*). CRISPR function in ATCC 4200RF was assessed with pKHS244, which contains the protospacer target for S244 present in the CRISPR1 array (*Figure 1a*). Conjugation frequencies of plasmids containing the protospacer were normalized to those of pKH12. We found that pKHS244 had a 56-fold lower acquisition rate than pKH12, and this phenotype was lost in ATCC 4200RF*Δcas9*, confirming that the reduction in conjugation frequency was due to CRISPR interference (*Figure 1b*). Additionally, we confirmed CRISPR1 activity in strains OG1RF and DS5 using pKHS96 and pKHS119, respectively (*Figure 1—figure supplement 2*). OG1RF is an oral commensal isolate that has been well characterized (*Dunny et al., 1978*; *Bourgogne et al., 2008*; *Gold et al., 1975*). DS5 is an erythromycin and tetracycline resistant isolate first described in 1974, and contains three extrachromosomal plasmids (*Clewell et al., 1974*) despite the fact that it possesses the entire CRISPR1-Cas system (*Palmer and Gilmore, 2010*). Our results demonstrate that a small conjugative vector can be used as an effective tool to evaluate the functions of individual spacers across a multitude of *E. faecalis* strains.

We have previously established that CRISPR1 and CRISPR2 are functionally linked in *E. faecalis* T11 (*Price et al., 2016*). We confirmed this in ATCC 4200RF with plasmid pKHS11, which is targeted by S11 on the ATCC 4200RF CRISPR2 locus. We observed a *cas9*-dependent decrease in conjugation frequency of pKHS11 relative to pKH12 (*Figure 1c*).

We then sought to examine if interaction between CRISPR1 Cas9 and CRISPR2 was preserved in MDR strains that natively lacked the CRISPR1-Cas system, since that had yet to be determined. We used the hospital-adapted strains V583 and Merz96, both MDR bloodstream infection isolates (*Sahm et al., 1989*; *Harrington et al., 2004*). Since these strains lack CRISPR1-Cas, *cas9* with its predicted promoter and predicted tracrRNA was introduced into a previously established chromosomal insertion site for expression in *E. faecalis* (*Price et al., 2016*; *Debroy et al., 2012*). The resulting strains were designated V649 and M236, originating from V583 and Merz96, respectively (*Figure 2a* and *Figure 2—figure supplement 1a*). We observed *cas9*-dependent decreases in conjugation frequencies of protospacer-bearing pKHSX plasmids relative to pKH12 for both V583 and Merz96 derivatives (*Figure 2b* and *Figure 2—figure supplement 1b*).

Previous transcriptional profiling performed in *E. faecalis* V583 (*Fouquier d'Hérouel et al., 2011*; *Innocenti et al., 2015*) identified bidirectional transcripts spanning the CRISPR2 region in V583. We independently confirmed the presence and start site of the transcript originating from the CRISPR2 leader region using primer extension (*Supplementary files 1* and *2*); we refer to this as the CRISPR2 transcript hereafter. The transcriptional start site we identified for the CRISPR2 transcript occurs at 85 bp upstream of the first nucleotide of the first CRISPR2 repeat, which agrees with previous findings (*Innocenti et al., 2015*). To determine if CRISPR2 transcription is required for genome defense, ~50 bp of its promoter was deleted from V649, creating the strain V254. V254 shows conjugation frequencies for pKHS67 similar to that of pKH12 (*Figure 2b*). Collectively, these data demonstrate that functional linkage of CRISPR2 and CRISPR1-Cas9 is preserved across diverse lineages of *E. faecalis*. Moreover, CRISPR2 can be activated for genome defense in *cas* deficient MDR strains of *E. faecalis*.

## Sequence-degenerated terminal repeats create functionally inactivated terminal spacers

In many CRISPR-Cas loci, the terminal CRISPR repeat is degenerated in sequence relative to the direct repeats (*Touchon and Rocha, 2010*; *Horvath et al., 2008*; *Almendros et al., 2014*). CRISPR2

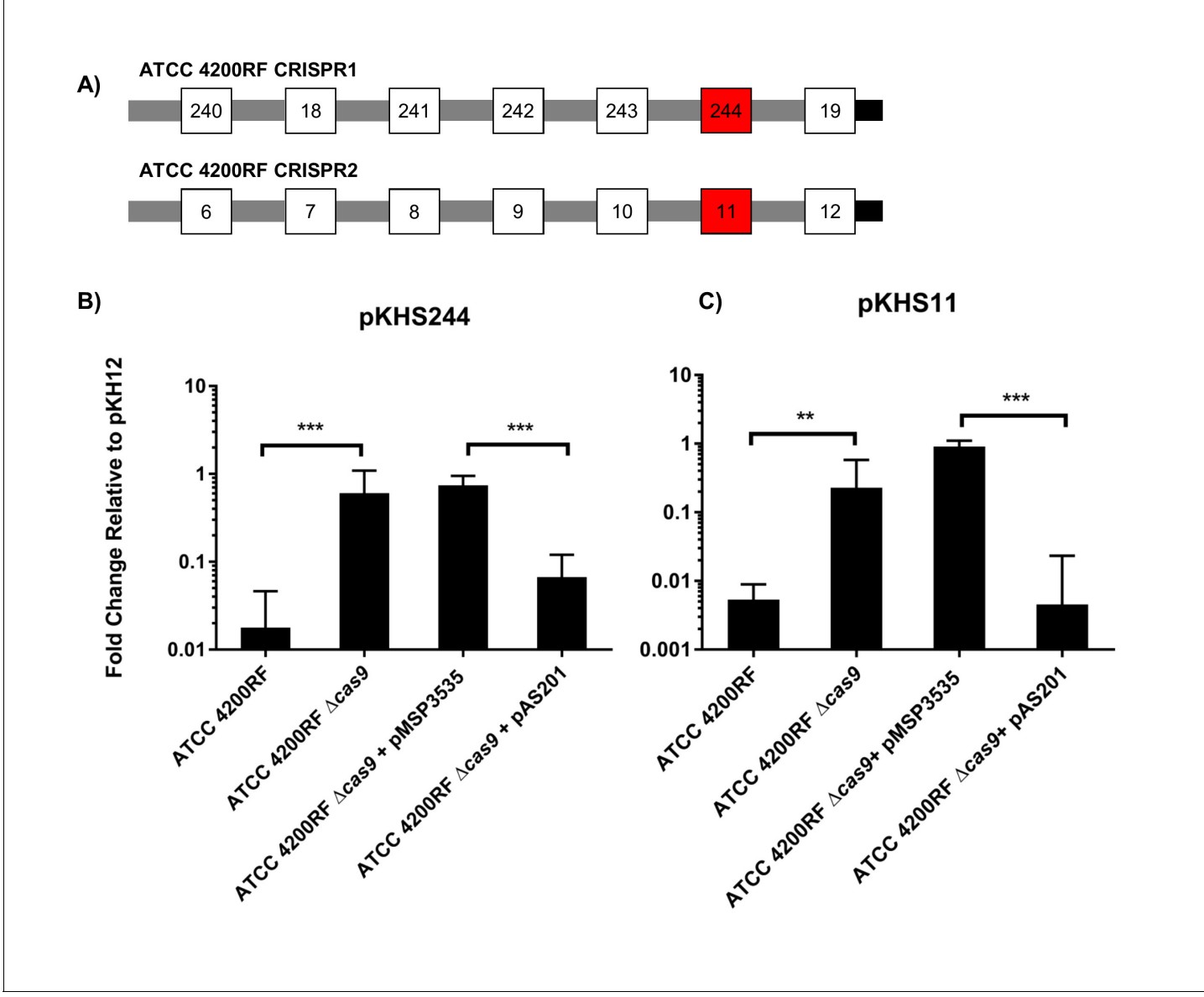

**Figure 1.** CRISPR defense in ATCC 4200RF. (**A**) ATCC 4200RF CRISPR1 and CRISPR2 arrays are shown. Spacers assessed for functionality in CRISPR1 or CRISPR2 are shown in red. Fold changes of transconjugants per donor relative to pKH12 are shown for CRISPR1 using pKHS244 (**B**) and CRISPR2 using pKHS11 (**C**). pAS201 encodes CRISPR1-*cas9* in a pMSP3535 backbone to complement the *cas9* deletion strain. Reactions containing pMSP3535 and pAS201 were performed on BHI with erythromycin, and C173 was the donor for all reactions. The geometric mean and geometric standard deviation of 5 and 3 independent biological replicates for CRISPR1 and CRISPR2, respectively, are shown. ***p<0.001, **p<0.01.

The following figure supplements are available for figure 1:

**Figure supplement 1.** CRISPR2 locus architecture and conjugation experiment setup.

**Figure supplement 2.** Conjugation assays to examine CRISPR1-Cas functionality in OG1RF and DS5.

terminal repeats completely degenerate after 19 bp, and the majority of arrays (including in V583) contain two polymorphisms within those 19 bp. These two polymorphisms are not observed in the majority of CRISPR1 terminal repeats, which degenerate after 22 bp (*Figure 2—figure supplement 2*). We hypothesized that the degeneracy of the terminal repeats of both CRISPR1 and CRISPR2 causes a loss of function in the terminal spacer, since the association of the crRNA with the tracrRNA

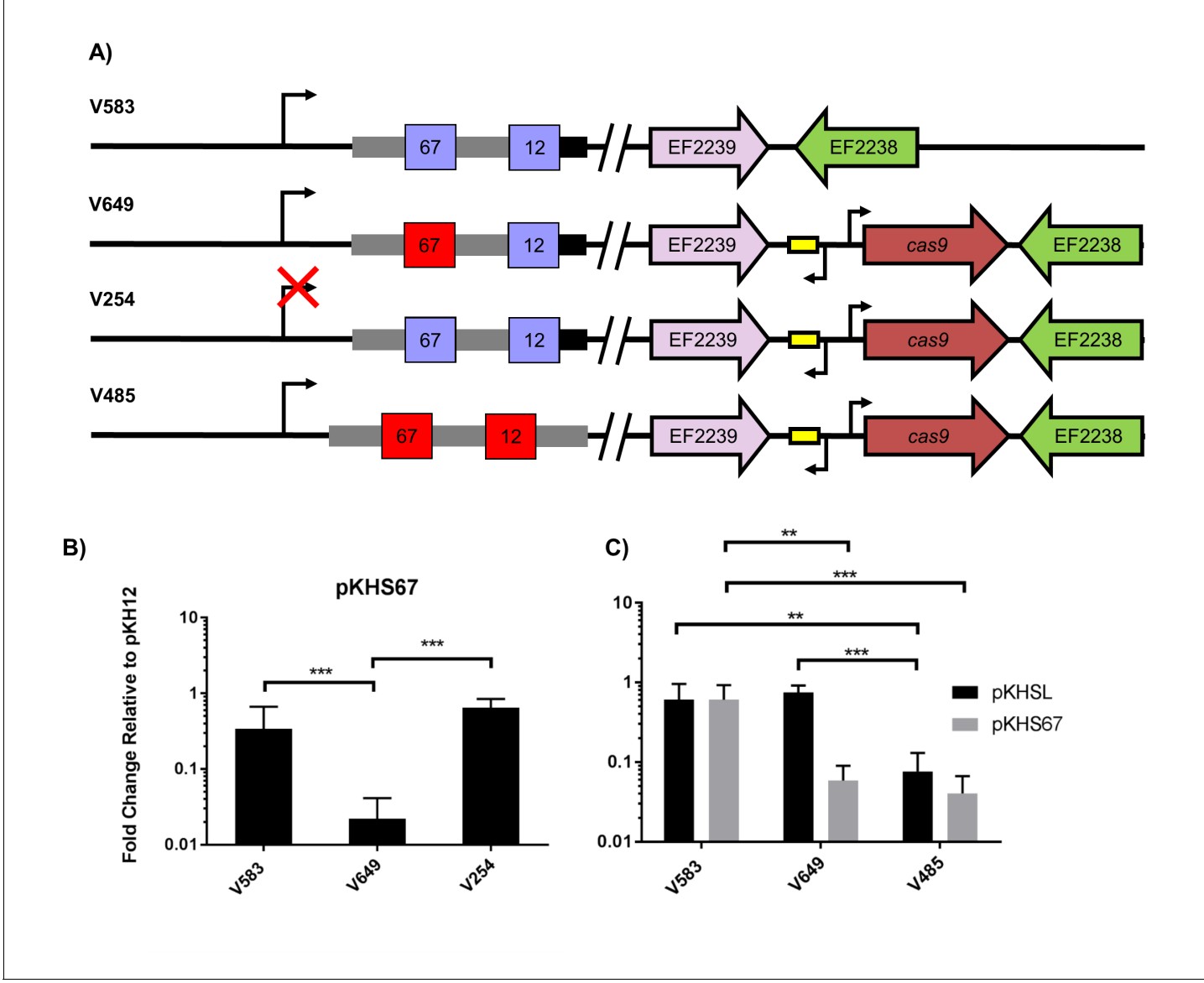

**Figure 2.** Assessment of CRISPR2 function in V583. (**A**) Genotypes of relevant strains are diagrammed. In V485, the CRISPR2 terminal repeat is replaced with a direct repeat. Spacers in red were experimentally determined to provide genome defense, while blue spacers do not provide genome defense. (**B**) Conjugation frequencies relative to pKH12 are plotted for pKHS67 transfer into V583, V649, and V254. The geometric mean and geometric standard deviation are shown for 4 independent biological replicates. (**C**) Conjugation frequencies relative to pKH12 are plotted for pKHSL and pKHS67 transfer into V583, V649, and V485. pKHSL contains the S12 sequence from the V583 CRISPR2 array. The geometric mean and geometric standard deviation for 3 independent biological replicates are shown. ***$p < 0.001$, **$p < 0.01$.

The following figure supplements are available for figure 2:

**Figure supplement 1.** CRISPR2 function in Merz96.

**Figure supplement 2.** Repeat alignments.

**Figure supplement 3.** Conjugation of plasmids containing terminal spacers are unaffected by deletion of *cas9*.

**Figure supplement 4.** Conservation of spacers across CRISPR2.

would be disrupted. V583 contains two spacers: S67, the target of our previous assay (see *Figure 2a*), and S12, which is associated with the terminal repeat. We first examined whether S12 was functional for genome defense. To avoid confusion, we named the engineered plasmid with the S12 protospacer target, pKHSL. As predicted, pKHSL conjugation frequency was equivalent to pKH12 in V649, revealing the lack of function of S12. This phenomenon was also apparent for terminal spacers in both the CRISPR1 and CRISPR2 arrays in ATCC 4200RF, as the deletion of *cas9* had no effect on the conjugation frequencies of plasmids bearing a terminal protospacer (*Figure 2—figure supplement 3a–b*). To confirm that this was due to the degenerated repeat, we replaced the terminal repeat in V649 with a direct repeat, creating the strain V485 (*Figure 2a*). In V485, S12 is active for interference (*Figure 2c*).

## Effect of CRISPR-plasmid incompatibility on maintenance of plasmids and spacers

Despite the fact that the intact CRISPR-Cas interference machinery in V649 targets pKHS67 for cleavage, we were still able to obtain transconjugants at approximately $\sim10^5$ CFU/mL, in agreement with previous findings for the native CRISPR3-Cas system of *E. faecalis* T11 (*Price et al., 2016*). This initially suggested that a large number of cells with CRISPR-inactivating mutations existed in the recipient population. If the CRISPR machinery was mutated before conjugation occurred, no change should occur in the maintenance of plasmids. To assess this, we conjugated pKHS67 into V583 and V649, and passaged two transconjugants each for 14 days. Cas9-dependent plasmid loss was observed in pKHS67 transconjugants in the absence of selection for the plasmid (*Figure 3*). CRISPR adaptation (i.e., new spacer addition) against pKHS67 was not observed. We infer that the CRISPR-Cas machinery was functional in the V649 transconjugants prior to conjugation.

We next examined the effect of forced maintenance of CRISPR targets over time. We monitored spacer deletion, as this is a rapidly screenable phenotype for variation in the CRISPR locus. Amplification of CRISPR2 from each passage revealed spacer loss events in 2 of 4 V649 pKHS67 transconjugants passaged with antibiotic selection for the plasmid (*Figure 4a*). The other 2 V649 pKHS67 transconjugants maintained pKHS67 with selection over the course of 14 days with no detectable spacer deletion events. These results are summarized in *Figure 4*. Importantly, we observe that *E. faecalis* cells can proliferate for days with their CRISPR-Cas system in conflict with a CRISPR target. Our results are in contrast with previous findings in other organisms, where the investigators describe the inability for CRISPR systems and their targets to co-exist in the same cell without mutations (*Lopez-Sanchez et al., 2012*; *Jiang et al., 2013a*; *Gomaa et al., 2014*; *Jiang et al., 2013b*). In *E. faecalis*, forced maintenance of a CRISPR target results in a fitness cost to the cell (*Figure 5—figure supplement 1*). We propose that, during passaging with or without selection, cells that are best adapted to overcome the CRISPR-induced fitness cost proliferate over time. When selection for the plasmid is present, the most competitive mutants have compromised the integrity of the CRISPR-Cas machinery likely through multiple mechanisms, one of which is spacer deletion. In the absence of selection, the fitness cost is overcome by plasmid loss.

## Terminal degenerate repeats prevent terminal spacer deletion in *E. faecalis* CRISPR2

The development of the spacer deletion assay allowed us to examine a largely unexplained characteristic of endogenous CRISPR systems: the role of terminal repeats and the conservation of terminal spacers (*Hullahalli et al., 2015*; *Horvath et al., 2008*; *Bachmann et al., 2014*; *Fabre et al., 2012*). To quantify this conservation, we returned to our previous analysis of CRISPR2 arrays in *E. faecalis* (*Hullahalli et al., 2015*). We generated a conservation index defined by the probability of a spacer in a particular position only existing in that position, taking into account 78 unique CRISPR2 arrays (*Figure 2—figure supplement 4*). Expectedly, the fractional conservation value of the terminal spacer is 0.999; that is, if a spacer is found in the terminal position, it is likely to only exist in the terminal position. Spacers upstream in the array show fractional conservation values of $\sim$0.5, meaning that there is a 50% probability that a spacer found upstream in the array will be found in the same position in a different array. Therefore, terminal spacers are the least likely to be lost from the CRISPR array.

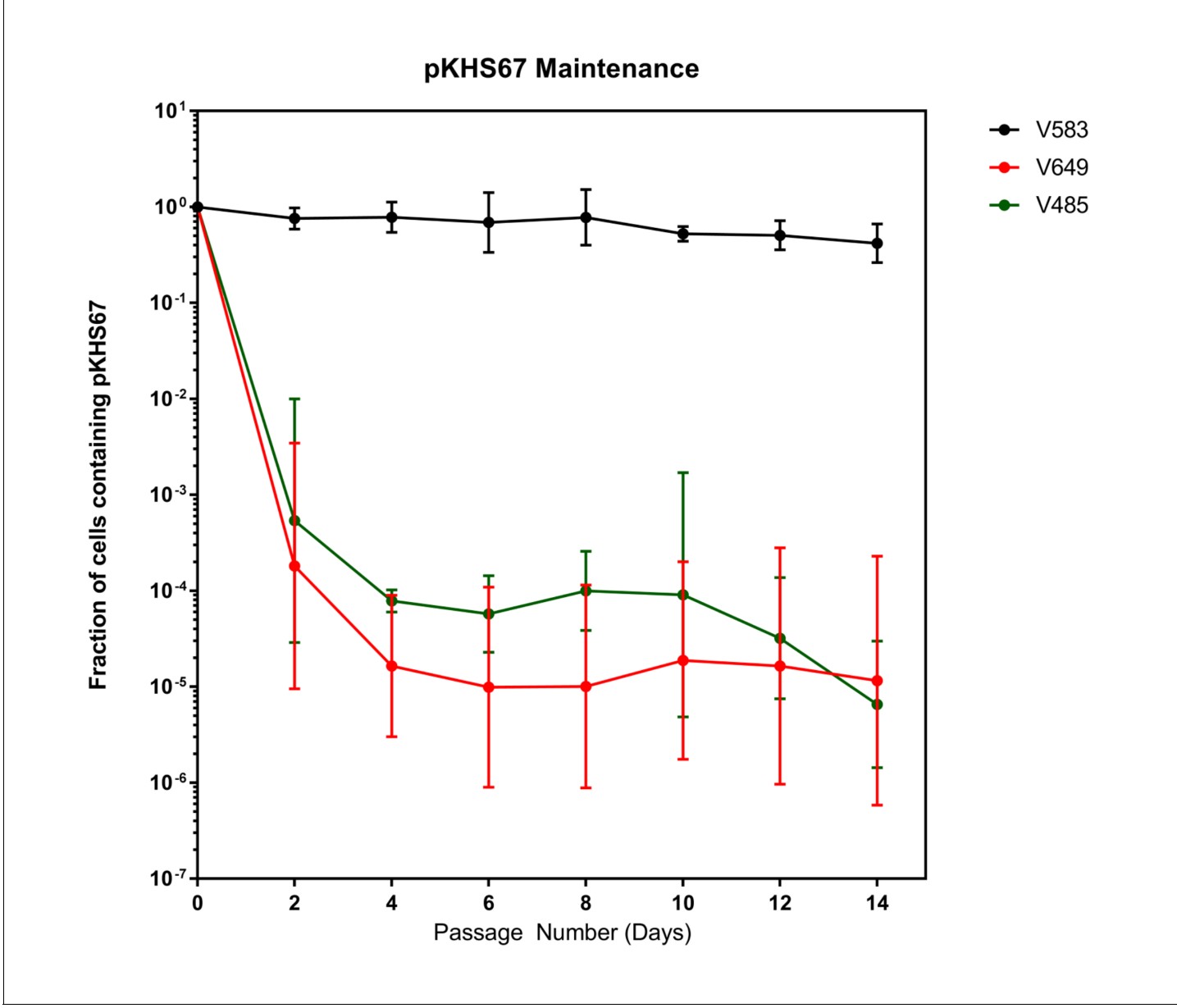

**Figure 3.** Plasmids are lost in a *cas9*-dependent manner in the absence of selection. Fraction of cells containing pKHS67 was determined by CFU-cam[R]/Total CFU. Cultures were passaged for 2 weeks in the absence of chloramphenicol. The geometric mean and geometric standard deviation from two independent biological replicates each from two transconjugants (total 4 unique transconjugants) are shown. Strains are defined in *Figure 3*.

Since we observe that terminal spacers are naturally non-functional for genome defense due to the degenerate downstream repeat (*Figure 2c*), it was of interest to determine if any evolutionary advantage is provided by the terminal degenerate repeats. A previous study proposed that terminal repeats act as 'anchor' units to preserve CRISPR integrity by preventing homologous recombination with upstream CRISPR repeat sequences (*Almendros et al., 2014*). Moreover, it is widely understood that terminal spacers correspond to the 'oldest' memories of the CRISPR array, since adaptation occurs at the leader end (*Barrangou et al., 2007*; *Sternberg et al., 2016*). In our spacer deletion assay, we observed only non-terminal spacer deletions when selection was applied to V649 pKHS67 transconjugants (*Figure 4*). We also passaged V485 pKHS67 transconjugants for 14 days, observing terminal and non-terminal spacer deletions in 2 of 3 cases. To increase the sample size, pKHS67 was conjugated into V649 and V485, and ten transconjugants each were passaged with

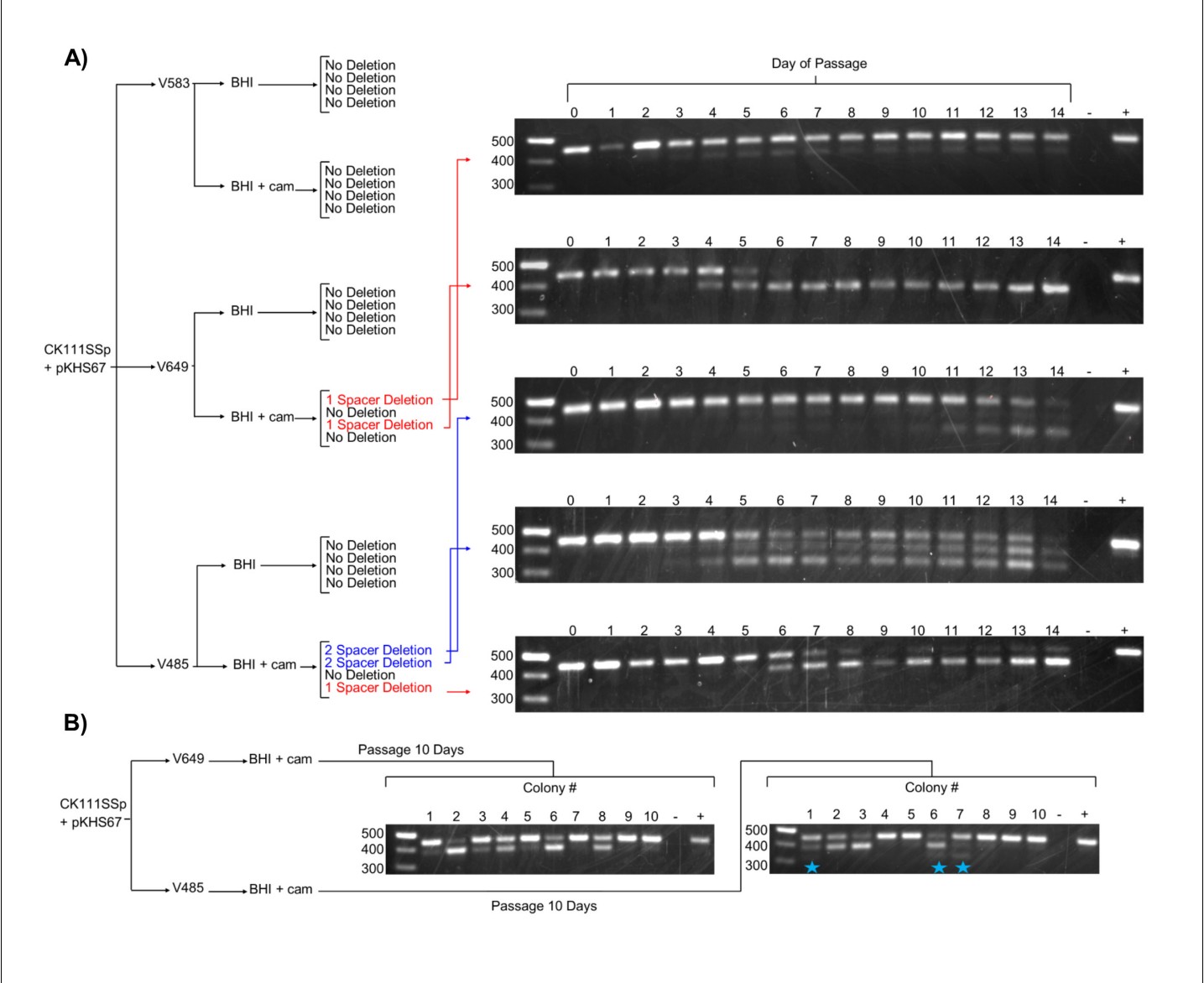

**Figure 4.** Effects of antibiotic selection on spacer content, and the role of terminal repeats. (A) Experimental schematic for spacer deletion assay. Transconjugants of V583, V649, and V485 with pKHS67 were passaged in the presence of chloramphenicol selection and CRISPR2 size was monitored across 14 days. Representative gel images are shown for all 14 days. −, negative control (water); +, positive control (V583 gDNA). (B) The experiment from (A) was repeated only for 10 transconjugants each of V649 and V485, and the CRISPR2 sizes from day 10 are shown for all 10 clones. Blue star indicates two-spacer deletions, which were only observable with V485.

selection for ten days. Three terminal spacer deletion events were observed in V485, while no terminal spacer deletion events were observed in V649 (*Figure 4b*). This observation provides evidence that terminal degenerate repeats in CRISPR2 prevent terminal spacer deletion and facilitate the persistence of the ubiquitous CRISPR2 array in *E. faecalis*. Additionally, this phenomenon helps explain the high degree of conservation of terminal spacers.

## CRISPR-plasmid incompatibility alters population structure

We observed that transconjugants with plasmids in conflict with CRISPR spacers exhibited an extended lag phase when grown with selection (chloramphenicol), while control transconjugants exhibited no such defect (*Figure 5—figure supplement 1*). We hypothesized that this growth

defect, in the presence of selection, could be exploited for the selective removal of strains based on CRISPR incompatibility. To test this, conjugation reactions containing mixed donors, recipients, and transconjugants were used as inocula for overnight cultures supplemented with chloramphenicol. This allows for a measurement of the competitive fitness of each strain when forced to maintain a plasmid. CFU counts of donors and transconjugants were conducted from the mating reaction itself and from each day of chloramphenicol passaging. Changes between the relative abundance of each strain before and after chloramphenicol passaging would reveal any inhibitory effects of forced maintenance of CRISPR targets in competitive environments relative to controls.

First, we examined the effects of using the donor strain CK111SSp(pCF10-101) as the competitor against V583 and V649 using a 1/1000 dilution of the conjugation reaction in BHI supplemented with chloramphenicol. We used pKH12 and pKHS67 to confer either no defense or CRISPR interference against the plasmid, respectively, in V649. When pKH12 and pKHS67 are conjugated into V583, V583 remains abundant in the population chloramphenicol passage (*Figure 5a–b*) and is ultimately able to outcompete CK111SSp(pCF10-101). Similarly, when pKH12 is conjugated to V649, CK111SSp(pCF10-101) is also ultimately outcompeted (*Figure 5c*). However, when pKHS67 is conjugated to V649, the V649 CFU is dramatically reduced in the population when passaged in chloramphenicol, while CK111SSp(pCF10-101) remains abundant (*Figure 5d*). Importantly, our data suggest that V583 and V649 naturally outcompete CK111SSp(pCF10-101) in BHI when no CRISPR conflict is present. However, when CRISPR conflict is applied, the opposite effect is seen. We also note that a 1/10,000 dilution of the mating reaction yields similar experimental results but requires fewer days of passaging (*Figure 5—figure supplement 2*), and this higher dilution was utilized for subsequent experiments. This demonstrates that competitive environments and CRISPR incompatibility can be used to remove certain MDR *E. faecalis* from a population in vitro.

To examine this effect on more complex systems, we conjugated pKH12, pKHS11, and pKHS67 from CK111SSp(pCF10-101) into ATCC 4200RF (possesses CRISPR1-Cas and S11) and either V583 or V649 (possess S67 but differ in *cas9* presence). As expected, when pKHS67 is propagated in the population, V649 falls to 0.001% of the population after the chloramphenicol passage. However, when pKHS11 is propagated, ATCC 4200RF falls below the threshold of detection, and V649 and CK111SSp(pCF10-101) dominate the population. It should be noted that in almost all experimental scenarios, V583 and V649 are more competitive than ATCC 4200RF. Only when challenged with CRISPR incompatibility by pKHS67 is V649 less competitive than ATCC 4200RF. Expectedly, V583 is never removed from the population since it lacks a functional CRISPR-Cas system. This experiment is summarized in *Figure 6*.

## CRISPR-mediated removal of antibiotic resistance genes

In natural *E. faecalis* populations, antibiotic resistance genes and virulence factors are often disseminated on PRPs, which are large mobilizable plasmids unique to the *E. faecalis* species, and which often encode toxin-antitoxin systems or bacteriocins to promote self-preservation (*Hirt et al., 2005*; *Weaver, 2012*; *Wardal et al., 2010*). Targeting PRPs may be a viable method to combat the spread of antibiotic resistance and virulence genes in *E. faecalis*. Previous studies have shown that the employment of self-targeting spacers is lethal to bacteria (*Jiang et al., 2013a*; *Gomaa et al., 2014*; *Selle et al., 2015*; *Citorik et al., 2014*; *Bikard et al., 2014*). However, in *E. faecalis,* we find that CRISPR systems and their targets can temporarily co-exist in the same cell, albeit at a fitness cost. We sought to capitalize on this and utilize CRISPR targeting in conjunction with competition to selectively deplete antibiotic resistance from enterococcal populations.

To engineer self-targeting constructs, we inserted the entire CRISPR2 locus from V583 into pKH12, generating pCR2. We then manipulated this locus to target two genes in V583: the *ermB* gene encoding erythromycin resistance located on the PRP pTEF1, and the EF0348 locus (encoding a putative phage tail protein) located on prophage 1 (*Paulsen et al., 2003*), generating pCR2-*ermB* and pCR2-Phage1, respectively. These plasmids were introduced into CK111SSp(pCF10-101) and conjugated into V583 and V649. Similar to native CRISPR-Cas defense, we observed a *cas9*-dependent reduction in pCR2-*ermB* and pCR2-Phage1 conjugation frequency relative to pCR2 (*Figure 7a–b*). However, what we term here as a reduction in 'conjugation frequency' should be further examined, since neither pCR2 derivative is targeted by the chromosomal CRISPR-Cas system. Rather, we hypothesize that the reduction in apparent conjugation frequency is due to growth inhibition in competitive mixtures. During initial growth of the conjugation reaction, cells that are subject to genomic

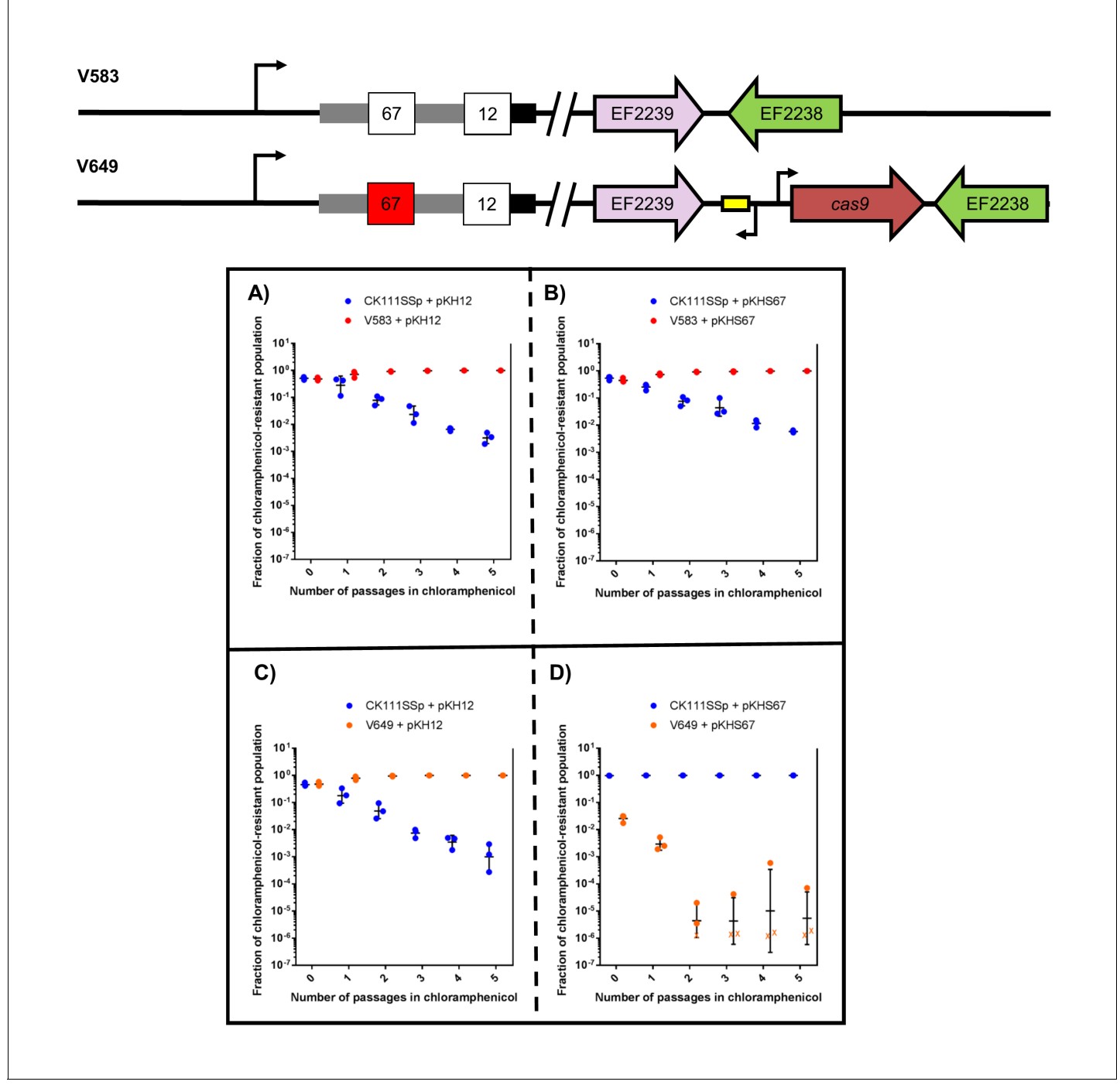

**Figure 5.** CRISPR alters population structure in vitro in a two-strain competition. Conjugation competition assays were performed with CK111SSp (pCF10-101) and V583 with (**A**) pKH12 and (**B**) pKHS67 using a 1/1000 dilution of the initial mating reaction for the first chloramphenicol passage. Experiments with CK111SSp(pCF10-101) and V649 were simultaneously performed with (**C**) pKH12 and (**D**) pKHS67. 'X' values indicate those that fell below the threshold of detection. Geometric mean and standard deviation are shown for 3 independent biological replicates. V583 and V649 genotypes have been shown again for clarity.

The following source data and figure supplements are available for figure 5:

**Source data 1.** Replicate values for growth curves.

**Figure supplement 1.** CRISPR targeting results in growth defect in chloramphenicol.

*Figure 5 continued on next page*

*Figure 5 continued*

**Figure supplement 2.** CRISPR alters population structure in vitro in a two-strain competition at higher initial dilutions.

'self-targeting' are less competitive, which allows surrounding cells (donors and plasmid-free recipients) to outcompete transconjugants during the course of the mating reaction. This results in significantly fewer transconjugants observed relative to controls. Nonetheless, our data show that targeting a native PRP in V649 was as detrimental to V649 as targeting a chromosomal region. This is presumably because pTEF1 loss activates toxin-mediated killing via its predicted toxin-antitoxin system (*Weaver, 2012*). Importantly, V649 transconjugants containing pCR2-*ermB* retained phenotypic resistance to erythromycin despite CRISPR targeting of *ermB* (data not shown). However, they exhibited a growth defect with chloramphenicol selection for pCR2-*ermB* (*Figure 5—figure supplement 1*).

We subsequently used pCR2-*ermB* to target pAM771; this plasmid is a non-cytolytic derivative of pAD1, a well-characterized PRP, which was obtained by mutagenesis with Tn*917* conferring erythromycin resistance via *ermB* (*Clewell, 2007*; *Ike and Clewell, 1984*). We generated pAM771 transconjugants in a CRISPR3 *cas9* deletion mutant of *E. faecalis* T11, and a derivative of this strain expressing CRISPR1-*cas9* and tracrRNA (referred to as T11CR1 [*Price et al., 2016*]). These *ermB*-encoding transconjugants were then used as recipients for conjugation with pCR2 and pCR2-*ermB*. We found that there was no reduction in conjugation frequency into these strains when transconjugants were selected with chloramphenicol (for pCR2 transconjugants) and rifampicin plus fusidic acid (for recipient cells). However, the number of transconjugants that retained erythromycin resistance was reduced 261-fold when pCR2-*ermB* was conjugated into T11CR1 (*Figure 7c*). We conclude that pTEF1 loss is not well tolerated by V649 but pAM771 loss is easily tolerated by T11CR1.

Finally, we evaluated the effect of targeting *ermB* in conjugation-competition assays, using CK111SSp(pCF10-101) as the donor and V649 (natively possesses pTEF1) and T11CR1 pAM771 as recipients (*Figure 8a*). When pCR2-*ermB* is propagated, erythromycin-resistant CFU decrease to 0.016% of the population, while the total V649 CFU falls below the threshold of detection after one passage in chloramphenicol. T11CR1 remains abundant, further emphasizing that loss of pAM771 is well tolerated by T11CR1. The total erythromycin resistant CFU is greater than total V649 CFU, indicating that residual erythromycin resistance in the population is due to pAM771. As expected, erythromycin resistance is not eliminated when pCR2 is propagated (*Figure 8b–c*). If T11CR1 pAM771 is excluded from the experiment (competition only with V649 and CK111SSp(pCF10-101)), total erythromycin-resistant CFU as well as total V649 CFU fall below the threshold of detection (*Figure 8—figure supplement 1*). Taken together, we conclude that pAM771, but not T11CR1, is suppressed by targeting *ermB* since loss of pAM771 is tolerated, while V649 is eliminated if pCR2-*ermB* is propagated since loss of pTEF1 is not tolerated. Our data indicate that CRISPR delivery via conjugation induces a fitness cost in the presence of selection that can be exploited in competitive environments to reduce the prevalence of specific genes and specific MDR strains from *E. faecalis* populations.

## CRISPR self-targeting does not induce *recA* expression in *E. faecalis*

We additionally sought to identify a potential mechanism by which CRISPR targets are able to be transiently maintained in *E. faecalis* despite active CRISPR-Cas systems. A previous study showed that expression of chromosomal CRISPR targeting in *E. coli* induced the SOS response, specifically activating *recA* (*Cui and Bikard, 2016*). We examined if chromosomal targeting by CRISPR-Cas induced a similar DNA damage response in *E. faecalis*. V649 pCR2 and pCR2-Phage1 transconjugants were harvested directly from transconjugant selection plates and used for RNA isolation. Subsequent culturing was avoided in order to limit the potential for CRISPR-related mutations that could accumulate over time. As a control, we induced the SOS response in *E. faecalis* cultures with the fluoroquinolone levofloxacin (LVX). Fluoroquinolones induce a canonical DNA damage response in enterococci and other bacteria (*Sinel et al., 2017*; *Baharoglu and Mazel, 2014*). We observed a 26-fold increase in *recA* transcript levels in cultures treated with LVX, while V649 pCR2-Phage1 transconjugants displayed a 1.7-fold decrease in *recA* transcript levels relative to V649 pCR2 transconjugants (*Figure 7—figure supplement 1a*). We simultaneously performed live/dead assays and found

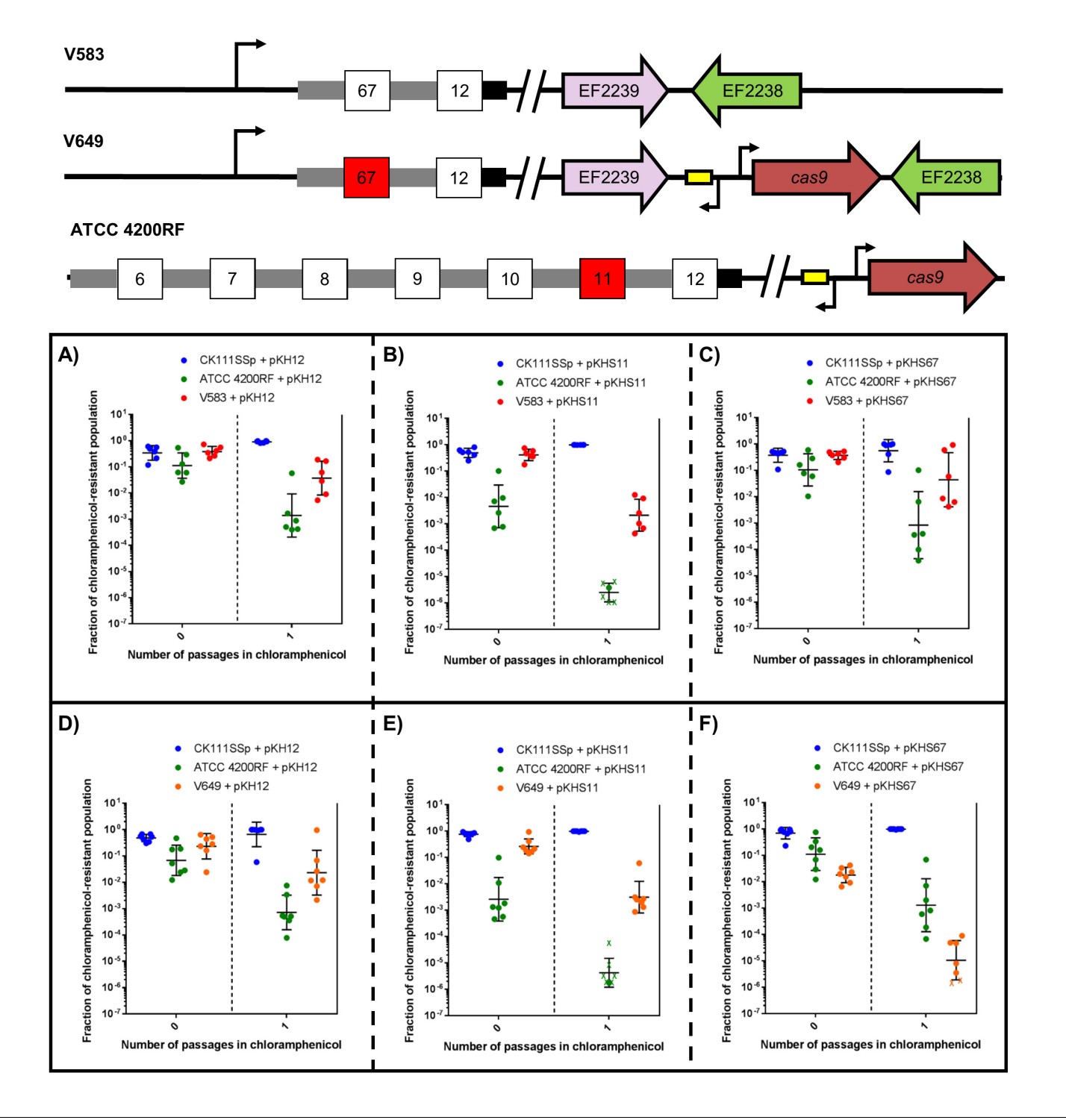

**Figure 6.** CRISPR defense alters population structure. The relative abundances of V583, V649, CK111SSp(pCF10-101), and ATCC 4200RF carrying a plasmid are plotted against the number of passages in chloramphenicol. Independent replicates are shown along with the geometric mean and geometric standard deviation. CK111SSp(pCF10-101), ATCC 4200RF, and V583 were propagated with (A) pKH12, (B) pKHS11 and (C) pKHS67. CK111SSp(pCF10-101), ATCC 4200RF, and V649 were also propagated with (D) pKH12, (E) pKHS11 and (F) pKHS67. The fraction of chloramphenicol resistant population was determined by strain-specific $cam^R$ CFU/total $cam^R$ CFU. 'X' indicate values that fell below the threshold of detection. Strain genotypes are shown for clarity.

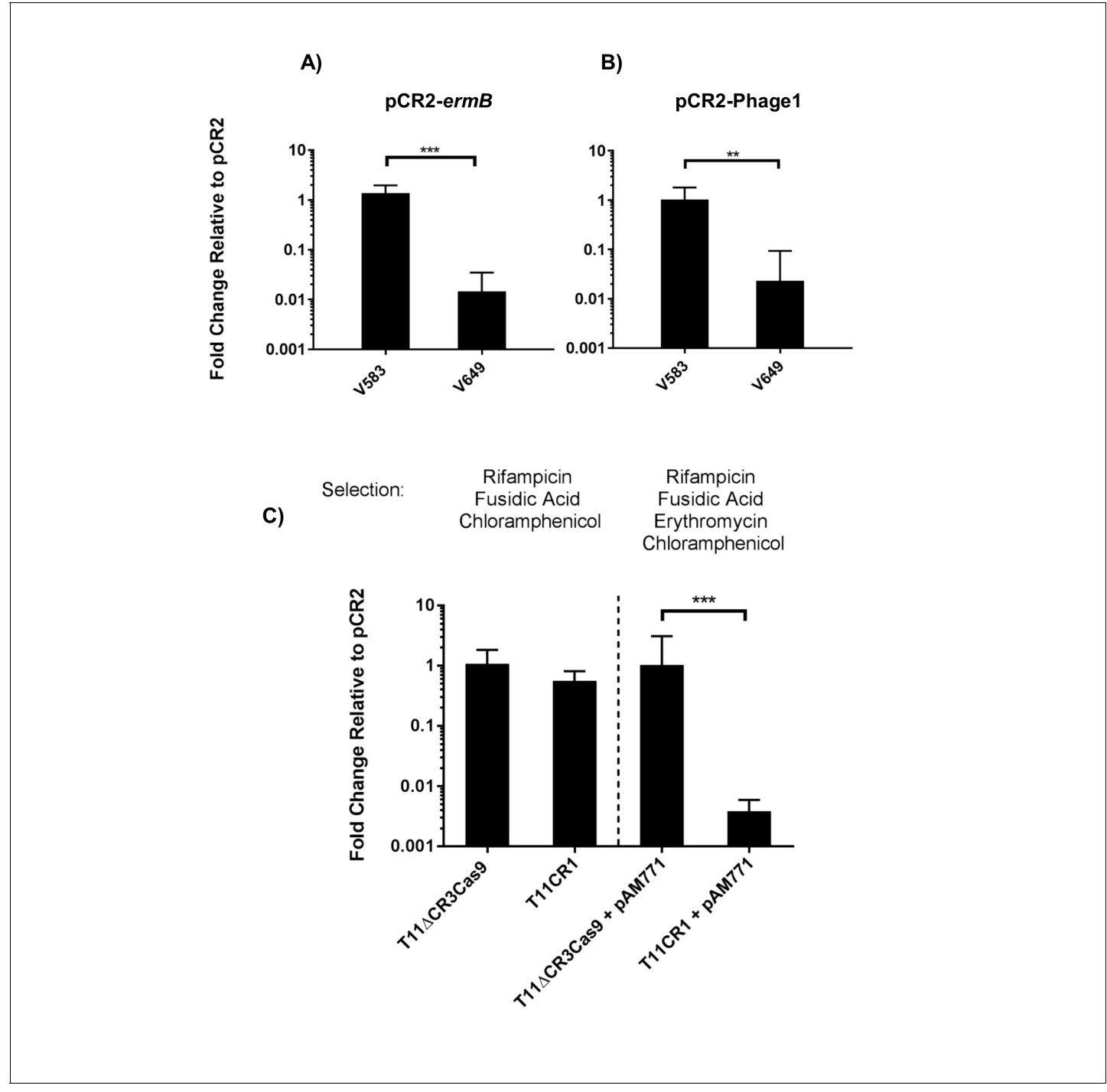

**Figure 7.** Transfer frequencies of 'self-targeting' plasmid constructs. (A) Conjugation frequency with pCR2-*ermB* are shown relative to pCR2 in V583 and V649. Transconjugants were selected on vancomycin and chloramphenicol. (B) Conjugation frequency with pCR2-Phage1 are shown relative to pCR2 in V583 and V649. (C) Conjugation frequency of pCR2-*ermB* are shown relative to pCR2 with varying transconjugant selection in T11RFΔCR3*cas9* and T11CR1. The geometric mean and geometric standard deviation of 3 independent biological replicates are shown. **p<0.01, ***p<0.001.

The following figure supplement is available for figure 7:

**Figure supplement 1.** RT-qPCR analysis of *recA* transcript levels and live cell quantification.

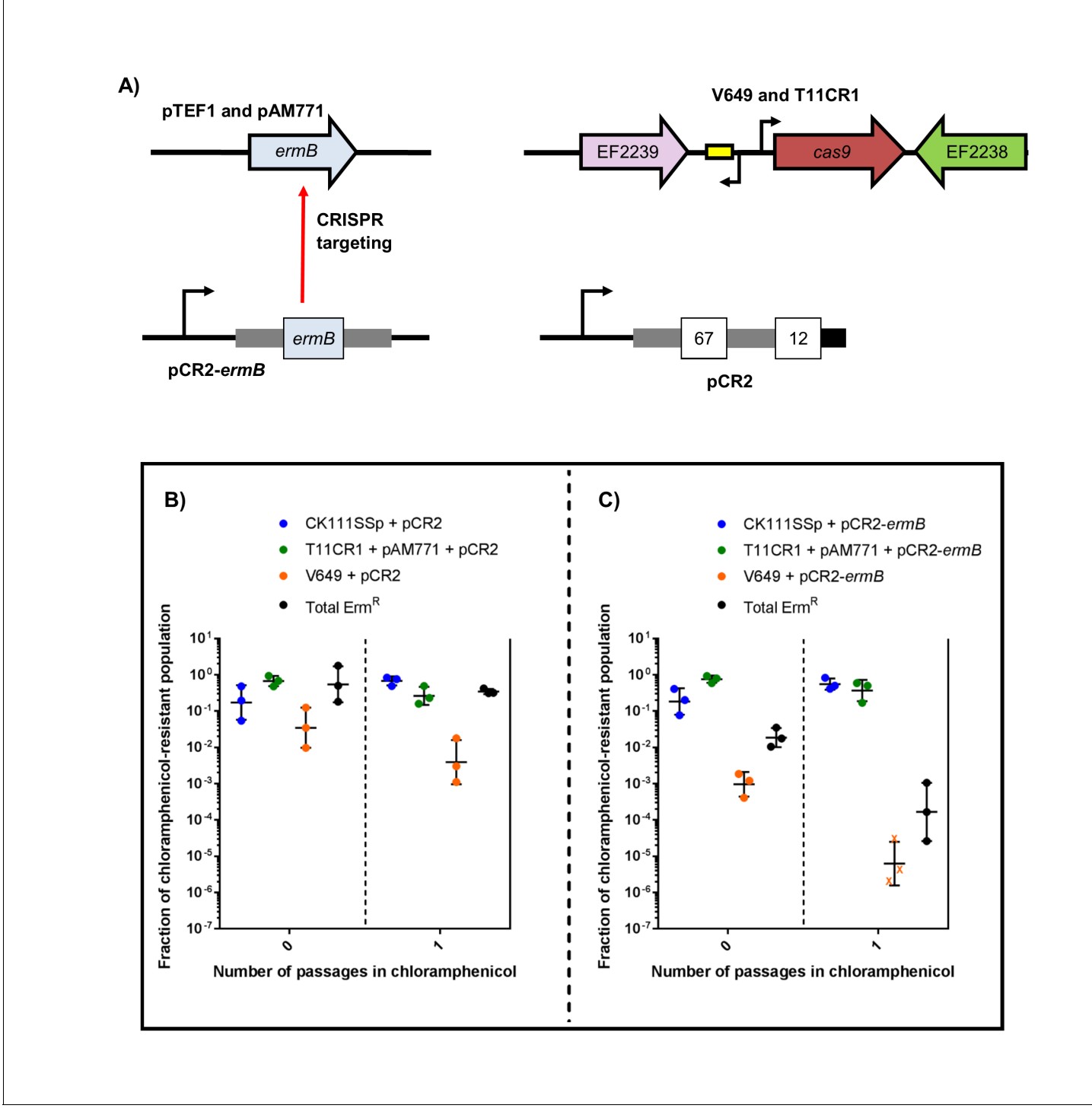

**Figure 8.** CRISPR targeting removes specific antibiotic resistances from populations in vitro. (**A**) Genotypes of select strains and plasmids are shown. (**B–C**) Conjugation-competition assays are shown, similar to those in *Figure 7*. T11CR1 carrying pAM771 was used, but transconjugants were selected on only rifampicin, fusidic acid, and chloramphenicol to assess if T11CR1 was able to survive with the loss of pAM771. V649 possesses *ermB* on pTEF1. Total erm[R] CFU accounts for both V649 and T11RF+pAM771. 'X' indicate values that fell below the threshold of detection.

The following figure supplement is available for figure 8:

**Figure supplement 1.** CRISPR targeting and two-strain competition can remove antibiotic resistances in vitro.

a small but statistically significant increase in cell death upon chromosomal CRISPR targeting (*Figure 7—figure supplement 1b*). These results suggest that CRISPR chromosomal targeting does not induce a canonical DNA damage response in *E. faecalis*.

## Discussion

In this study, we activated the orphan CRISPR2 locus of MDR *E. faecalis* for anti-plasmid genome defense. However, a large transconjugant pool still arises despite active CRISPR defense. Over time, these transconjugants lose plasmids when selection is absent and can lose spacers if selection is applied. Using this spacer loss assay, we show that CRISPR2 terminal repeats prevent terminal spacer loss, suggesting that CRISPR systems have an intrinsic mechanism for self-preservation. Even though plasmids can evade CRISPR interference at a high frequency, this limitation is overcome by intraspecies competition. *E. faecalis* strains exhibit growth defects when forced to maintain conflicts between CRISPR-Cas and its targets. Exploiting this phenomenon, we show that CRISPR-Cas can be used to deplete specific strains or antibiotic resistance genes from a population in vitro. The ability for CRISPR-assisted *E. faecalis* population control remains to be assessed in vivo. Future studies will seek to optimize the current system in vivo.

Terminal degenerate repeats and highly conserved terminal spacers are a common characteristic of many CRISPR systems. It has been previously proposed that the terminal degenerate repeats 'anchor' the CRISPR array (*Almendros et al., 2014*). We provide evidence that CRISPR2 terminal repeats can prevent terminal spacer deletion, but the evolutionary advantage remains unclear, especially since it has been shown that a single direct repeat alone is sufficient to promote spacer adaptation in *E. coli* and *S. thermophilus* (*Yosef et al., 2012*; *Wei et al., 2015*). In agreement with our findings, the smallest CRISPR2 array observed in *E. faecalis* consists of a direct repeat followed by a single spacer and a terminal repeat (*Hullahalli et al., 2015*). Degeneracy in the terminal repeat may have been selected over time, preventing complete loss of the CRISPR array, and as a consequence, the terminal spacer lost CRISPR interference functionality. Moreover, that terminal spacers are functionally inactive for genome defense means that they are not subject to conflicts with protospacer-bearing elements, which may also explain their conservation.

A particularly striking feature of CRISPR systems in *E. faecalis* is the ability of the targeted DNA to temporarily coexist with the CRISPR-Cas system. It has been previously shown that CRISPR-Cas systems and their targets cannot coexist in the same cell. This phenomenon has been observed in *E. coli, Sulfolobus,* and various streptococci (*Jiang et al., 2013a*; *Gudbergsdottir et al., 2011*; *Bikard et al., 2012*; *Marraffini and Sontheimer, 2010*; *Selle and Barrangou, 2015*). In a thorough investigation of CRISPR-Cas in *S. epidermidis*, Jiang, et al detected transconjugants at very low frequencies, and found that they all contained mutations that inactivated the CRISPR system (*Jiang et al., 2013b*). Similarly, a dramatic inhibition of growth was observed when CRISPR self-targeting was induced in *Pectobacterium atrosepticum,* which led to growth inhibition and cellular elongation and filamentation, which are indicators of the SOS response (*Vercoe et al., 2013*). CRISPR-induced toxicity was also found to induce the SOS response in *E. coli* at low *cas9* expression levels (*Cui and Bikard, 2016*). This is in direct contrast to our findings, where CRISPR conflict does not result in *recA* up-regulation. However, *Pseudomonas aeruginosa* displays characteristics of CRISPR interference similar to that of *E. faecalis*, where a targeted plasmid is acquired, but gradually lost from the population over time due to CRISPR interference (*Høyland-Kroghsbo et al., 2017*). Similar to our observations in *E. faecalis*, strikingly large numbers of 'escape mutants' were observed for *P. aeruginosa*. This delayed activity was attributed to the ability for plasmid replication to titrate out Cas9 targeting complexes. Our study is distinct in that we include chromosome-targeting constructs, which would not be limited by plasmid replication, and find that similarly large transconjugant pools appear. This suggests that plasmid copy number titrating Cas9 targeting complexes does not completely explain the transient tolerance of CRISPR targets.

Previous analysis of native *E. faecalis* CRISPR3-Cas also described a large number of purported 'escape mutants' that arose despite CRISPR defense (*Price et al., 2016*); the mechanism of escape was not investigated in that study. Our data suggest that these transconjugants may better be classified as 'unstable'. Based on the conditions subsequently applied, the cells stabilize themselves by adapting to the new environment, resulting in either CRISPR loss or plasmid loss. The basis for these transconjugants may be efficient DNA repair or stress response mechanisms that allow cells to

survive chromosome cleavage by Cas9, or the induction of genes that transiently repress the CRISPR system, but this is speculative. It is also possible that this phenotype is due to low native expression of *cas9*. If this were the case, it would suggest that *E. faecalis* has evolved to temporarily tolerate CRISPR targets, perhaps as a mechanism to enhance population-level genetic diversity. This could prove beneficial, particularly in environments where acquisition of a mobile genetic element is advantageous for survival. Further studies, such as RNA sequencing, are required to elucidate the mechanism of *E. faecalis* tolerance to CRISPR-Cas toxicity. It will also be of interest to use deep sequencing to fully elucidate the evolutionary outcomes of CRISPR-protospacer conflicts in *E. faecalis* populations and provide greater resolution beyond the spacer deletion assays used here.

An application for CRISPR-mediated toxicity is in the selective depletion of hospital-adapted strains from heterogeneous populations. CRISPR targeting to distinguish very closely related strains was first demonstrated in *E. coli* (*Gomaa et al., 2014*). Delivery of CRISPR-Cas systems is a major challenge to establishing CRISPR therapeutics, and has been studied using bacteriophage (*Citorik et al., 2014*; *Bikard et al., 2014*; *Yosef et al., 2015*). However, a major limitation to phage delivery of CRISPR systems is narrow host range and incomplete delivery (*Beisel et al., 2014*). Moreover, *E. faecalis* has been shown to readily acquire resistance to phage via receptor mutations (*Duerkop et al., 2016*). Conjugative delivery has been previously examined in *E. coli* (*Citorik et al., 2014*), but the investigators found very few viable transconjugants when introducing plasmids targeting the host chromosome via CRISPR. This is in contrast to what we observe in *E. faecalis*, where a large pool of transconjugants readily emerges despite CRISPR-Cas activity. We can exploit this by manipulating intra-species competition and CRISPR-Cas interference to deplete certain *E. faecalis* genes and strains from mixed populations. In this study, we used antibiotic selection to provide a competitive advantage to strains that readily accepted and maintained the plasmid. We showed that *E. faecalis* populations can be altered in vitro using this strategy. CRISPR-Cas may be a viable tool to alter in vivo E. faecalis populations as well. To deploy this system in vivo, however, it will be more feasible to utilize naturally occurring conjugative vectors such as the PRPs. PRPs have previously been shown to provide selective advantages in vivo through bacteriocin production, mimicking the antibiotic selection used in this study (*Kommineni et al., 2015*). Future studies will test the abilities of CRISPR constructs deployed on PRPs conferring beneficial competitive traits such as bacteriocins to manipulate *E. faecalis* populations in vitro and in vivo.

The use of CRISPR-Cas a tool to modify population structure is extremely attractive, due to its highly specific activity against particular strains, leaving the remainder of the microbiome intact. As demonstrated by our in vitro work, CRISPR-Cas employed on conjugative vectors that target antibiotic resistance genes can actively suppress proliferation of other conjugative vectors that harbor the targeted genes. Furthermore, drug-sensitive *E. faecalis* are already used as commercial probiotics (*Neuhaus et al., 2017*; *Domann et al., 2007*; *Fritzenwanker et al., 2016*). The use of CRISPR targeting constructs in combination with *cas9* to influence enterococcal populations, perhaps using probiotic *E. faecalis* armed with narrow host range conjugative delivery vectors, could be helpful in reducing MDR enterococcal bloodstream infections in patients colonized with these organisms (*Al-Nassir et al., 2008*; *Ubeda et al., 2010*; *Clutter et al., 2013*; *Rice, 2013*).

## Materials and methods

### Bacterial strains, growth conditions, and routine molecular biology procedures

*E. coli* strains EC1000 (*Leenhouts et al., 1996*) and JM109 were routinely cultured in LB at 37°C and shaken at 220 rpm for broth cultures, unless otherwise mentioned. All *E. faecalis* strains were routinely cultured in Brain-Heart Infusion (BHI) broth or agar at 37°C unless otherwise mentioned. Antibiotics were used in the following concentrations: chloramphenicol (cam), 15 µg/mL; erythromycin (erm), 50 µg/mL; vancomycin, 10 µg/mL; streptomycin, 500 µg/mL; spectinomycin, 500 µg/mL, rifampicin, 50 µg/mL; fusidic acid, 25 µg/mL, ampicillin (for *E. coli*), 100 µg/mL. Routine PCR for screening and sequencing was performed with *Taq* DNA Polymerase (New England Biolabs, Ipswich, USA). For cloning applications, Q5 DNA Polymerase (New England Biolabs) or Phusion DNA Polymerase (Fisher, Waltham, MA) were used. Calf Intestinal Phosphatase and Polynucleotide Kinase (New England Biolabs) were used as routine phosphatases and kinases as per manufacturer's instructions.

Restriction enzymes were acquired from New England Biolabs. Genomic DNA was isolated for routine PCR using the MO BIO Microbial DNA Isolation Kit. Plasmids and PCR products were purified with the GeneJet Plasmid Purification Kit (Fisher) and the PureLink PCR Purification Kit (Fisher). Sanger sequencing to confirm all genetic manipulations of bacterial strains and plasmids was carried out at the Massachusetts General Hospital DNA Core Facility. DNA oligos were synthesized by Sigma-Aldrich (St. Louis, MO). Routine plasmid cloning hosts were *E. coli* EC1000 or JM109 (for pGEM). JM109 chemically competent cells, EC1000 electrocompetent cells, and *E. faecalis* electrocompetent cells were prepared using previously published protocols (*Bhardwaj et al., 2016*), except that *E. faecalis* cell pellets were resuspended in 0.5 mL of lysozyme buffer instead of 10 mL. A full list of strains used is shown in *Supplementary file 3* and primers used in this study are listed in *Supplementary file 4*. Parental bacterial strains and plasmids were obtained from our prior studies or from colleagues in the field (see *Supplementary file 3* for more information). Genome sequence is publicly available (https://www.ncbi.nlm.nih.gov/genome/) for all parental *E. faecalis* strains used in this study.

## Generation of pKH and pCR2 derivatives to assess CRISPR function in *E. faecalis*

The *oriT* region of pHA101 (*Bhardwaj et al., 2016*) was cloned into the SalI restriction site of pLZ12 (*Perez-Casal et al., 1991*). The resulting plasmid was designated pKH12, and was confirmed as being able to conjugate from CK111SSp(pCF10-101) donors into a variety of recipients at high frequencies. pKH12 was linearized via PCR with primers that contained the protospacer and NGG PAM sequences, and the linear plasmid was phosphorylated and self-ligated to generate pKHSX, where X defines a spacer from our dictionary (*Hullahalli et al., 2015*). All pKHSX plasmids were propagated in EC1000 and sequenced prior to introduction into CK111SSp(pCF10-101) for conjugation experiments. To generate pCR2, we linearized pKH12 in a similar scheme as generating pKHSX derivatives and inserted the CRISPR2 locus of V583. Modifications to pCR2 were created by linearizing pCR2 around the CRISPR2 locus (immediately adjacent to the first repeat and terminal repeat) and inserting a repeat-spacer-repeat unit containing spacers targeting *ermB* and Phage1. All protospacer and spacer sequences with adjacent regions were sequenced-verified and possess 100% identity to the cognate spacer/protospacer and PAM. The sequence of pCR2 has been deposited in Genbank under the accession number MF157411.

## Plasmid construction for genome manipulation

Plasmids used to modify the *E. faecalis* chromosome (with the exception of CK111SSp(pCF10-101)) were derived from pLT06 or pHA101 (*Bhardwaj et al., 2016*; *Thurlow et al., 2009*). Plasmid pVP31, used to delete CRISPR1-*cas9*, was created by ligating XmaI/XbaI digested pLT06 to appropriate fragments upstream and downstream of *cas9* in ATCC 4200RF. Plasmid pG19 was generated previously (*Huo et al., 2015*; *Price et al., 2016*). pMR23, used to chromosomally integrate the wild-type CRISPR2 locus of V583 in its native position, was created by ligating BamHI-digested pHA101 with a fragment containing the V583 CRISPR2 locus with approximately 1 kb upstream and downstream segments. Plasmid pMR28, used to create a CRISPR2 promoter knockout in V583 derivatives, was generated by SphI ligation of fragments conferring the promoter deletion that were subsequently cloned into the EcoRI site of pHA101. To replace the terminal repeat with a direct repeat, pMR23 was linearized around the terminal repeat and two phosphorylated 36 bp direct repeat oligos were annealed and blunt-end ligated. All plasmids described in this section were mated into the desired strains from CK111(pCF10-101) derivatives, with the exception of pVP31 and pG19, which lack the *oriT* sequence necessary for conjugation. Electroporation was performed instead for pVP31 and pG19. To complement the CRISPR1-*cas9* deletion in trans, we cloned *cas9* into pMSP3535 (*Bryan et al., 2000*) at PstI/XhoI. The resulting plasmid was named pAS201.

Since CK111SSp(pCF10-101) encodes *repA* on the chromosome, it cannot be modified using derivatives of the previously mentioned plasmids, which rely on a temperature sensitive *repA* to create chromosomal modifications. To inactivate the native CK111SSp(pCF10-101) CRISPR1 system and simultaneously introduce erythromycin resistance, we created plasmid pGE17 using the HiFi DNA Master Mix (New England Biolabs). pGE17 contains the pGEM origin of replication, and appropriate arms and insert to introduce *ermB* to interrupt the *cas9* locus. Additionally, *pheS\** and *cat* from

pLT06 were included outside the homologous arms to provide appropriate selection and counter-selection. A full list of plasmids used in this study are shown in *Supplementary file 3*.

## Creation of chromosomal modifications

Modifications to *E. faecalis* strains other than CK111SSp(pCF10-101) were performed as previously described, using *para*-chloro-phenylalanine counterselection coupled with temperature shift (*Thurlow et al., 2009*; *Kristich et al., 2005*). To generate the insertion of *ermB* into the *cas9* locus of CK111SSp(pCF10-101), plasmid pGE17 was electroporated in large quantities (>2.5 µg) into CK111SSp(pCF10-101) and plated on BHI supplemented with chloramphenicol. Since pGE17 cannot extra-chromosomally replicate in *E. faecalis*, transformants were integrants that were resistant to both chloramphenicol and erythromycin, and usually appeared within 3–5 days. These integrants were inoculated in media containing both antibiotics (to make stocks and confirm integration), and subsequently passaged in BHI supplemented with only erythromycin. The resulting culture was diluted and plated on MM9YEG containing *p*-Cl-phenylalanine to select recombinants that lost the plasmid backbone. Erythromycin-resistant colonies that were chloramphenicol-sensitive were sequence confirmed as having the *ermB* cassette interrupting *cas9*.

## Conjugation and conjugation-competition assays

1 mL of overnight (~18 hr) cultures of donors and recipients were each washed in 1X PBS to remove excess antibiotic and media, and subsequently added to 5 mL of BHI containing no antibiotics. Cultures were incubated for 1.5 hr, after which 100 µL of the donor strain was mixed with 900 µL of the recipient strain, spread on BHI agar, and incubated overnight. Lawns were then scraped in PBS and dilutions were inoculated on plates containing appropriate antibiotics. Selection plates were subsequently incubated at 37°C or 30°C (for plasmids containing r*epA*^ts^). For experiments assessing CRISPR function, independent tests were performed to verify that cross-resistance to the selection for donors or recipients did not occur.

To evaluate the effects of CRISPR systems in competitive environments, the conjugation reaction obtained after plate scraping was diluted 1000-fold or 10,000-fold in BHI with chloramphenicol and incubated at 37°C overnight. CFU for each strain containing the plasmid were measured by plating on BHI containing chloramphenicol and appropriate antibiotics to differentiate unique strains. For reactions with two recipients, a 'Donor: Recipient 1: Recipient 2' ratio of 1:4.5:4.5 was used.

## Evolution assay

For each transconjugant colony assessed, half of the colony was inoculated into BHI, and half into BHI supplemented with chloramphenicol. Cultures were passaged daily for up to 2 weeks using 1:1000 dilutions from overnight cultures. Each day, the size of the CRISPR2 locus was monitored via PCR using primers CR2 Seq For and CR2 Rev (*Supplementary file 3*). Additionally, dilutions were plated on BHI and BHI supplemented with chloramphenicol to measure the retention of plasmids within the population.

## Statistical analysis

In a previous study (*Hullahalli et al., 2015*), we examined CRISPR2 arrays across 228 previously sequenced *E. faecalis* isolates. Filtering out redundant CRISPR2 arrays and unique spacers that occurred only once in the dataset, we calculated the frequency at which a particular spacer appears in a given position in the array, as a fraction of the total number of occurrences within the dataset. Each frequency value was given proportional weight to the number of times a spacer appeared within a specific position. The fractional conservation value for a particular position in the array was defined as the average of the weighted frequencies for all spacers that occur at a particular position. Statistical significance for conjugation assays was assessed using one-tailed student's T test assuming log-normal distribution.

## Primer extension

20 µg of DNA-free log-phase V583 RNA was used as a template for 6-FAM labeled primer extension as previously described (*Lloyd et al., 2005*). Reverse transcription was carried out using SuperScript II (Fisher) and cDNA was precipitated with ethanol and sent to the Oklahoma University Health

Sciences Center DNA core facility for determination of the size of the transcript. The start site of transcription was determined using Peak Scanner (Thermo Fisher). The results of two independent experiments verified the CRISPR2 start site.

## Growth curves

Individual transconjugant colonies were suspended in PBS and used to inoculate 96-well plates with or without chloramphenicol selection. Optical density was monitored for 24 hr using a microplate reader (BioTek Synergy).

## Quantitative PCR (RT-qPCR)

V649 pCR2 and V649 pCR2-Phage1 transconjugants were selected on vancomycin and chloramphenicol. After two days incubation, colonies were pooled without passaging (to prevent accumulation of CRISPR-related mutations). Total RNA was harvested using RNA-Bee and chloroform precipitation essentially as previously described (*Bhardwaj et al., 2016*). RNA was treated with DNase I (Roche) and concentrated using the GeneJet RNA Cleanup and Concentration Kit (Fisher). cDNA was subsequently synthesized using qScript cDNA Supermix (Quanta Biosciences). As a control, RNA was similarly harvested from log phase V649 cultures ($OD_{600nm}$ = 0.3) and two hours after addition of 1 μg/ml levofloxacin. qPCR was performed using the AzuraQuant Green Fast qPCR Mix (Azura Genomics) with primers qrecA for/rev and qclpX for/rev using the MyGo Pro qPCR machine (Phenix). The *clpX* transcript was used to normalize samples. Fold changes are expressed as $2^{\Delta\Delta Cq}$, where $\Delta\Delta Cq$ = $(Cq_{pCR2-Phage1}clpX - Cq_{pCR2-Phage1}recA) - (Cq_{pCR2} clpX - Cq_{pCR2} recA)$ or $(Cq_{LVX} clpX - Cq_{LVX} recA) - (Cq_{control} clpX - Cq_{control} recA)$. Both experiments (assessing CRISPR and LVX) were performed in biological triplicate.

## Live/dead staining

V649 pCR2 and V649 pCR2-Phage1 transconjugants were selected on vancomycin and chloramphenicol. After two days incubation, colonies were scraped and washed twice with 0.85% NaCl. Live/dead staining was performed using BacLight Bacterial Viability Kit (Fisher) per the manufacturer's instructions. Fluorescence was measured using a microplate reader (Biotek Synergy). Samples from three biological experiments were assayed in technical triplicates.

## Acknowledgements

This work was supported by R01 AI116610 to KLP and the American Society for Microbiology Undergraduate Research Fellowship to KH. We thank members of the Palmer Lab for critical feedback on the manuscript. We thank Ardalan Sharifi for the plasmid pAS201 and Valerie J Price for providing the strain ATCC4200RF*Δcas9*. We thank Dr. Michael Gilmore, Dr. Gary Dunny, and Dr. Breck Duerkop for providing strains and plasmids used in this study.

## Additional information

### Funding

| Funder | Grant reference number | Author |
| --- | --- | --- |
| National Institutes of Health | R01 AI116610 | Kelli L Palmer |
| American Society for Microbiology | Undergraduate Research Fellowship | Karthik Hullahalli |

The funders had no role in study design, data collection and interpretation, or the decision to submit the work for publication.

### Author contributions

KH, MR, Conceptualization, Formal analysis, Validation, Investigation, Methodology, Writing—original draft; KLP, Conceptualization, Formal analysis, Supervision, Funding acquisition, Project administration, Writing—review and editing

Author ORCIDs

Karthik Hullahalli, http://orcid.org/0000-0003-3064-2090

Marinelle Rodrigues, http://orcid.org/0000-0002-5509-605X

Kelli L Palmer, http://orcid.org/0000-0002-7343-9271

## Additional files

### Supplementary files

• Supplementary file 1. Primer extension replicate 1. Raw data for identification of the CRISPR2 start site is provided.

• Supplementary file 2. Primer extension replicate 2. Raw data for identification of the CRISPR2 start site is provided.

• Supplementary file 3. Strains and plasmids used in this study.

• Supplementary file 4. Primers used in this study.

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
