## [Decision Letter]

Thank you for submitting your article "Exploiting CRISPR-Cas to manipulate *Enterococcus faecalis* populations" for consideration by *eLife*. Your article has been favorably evaluated by Gisela Storz (Senior Editor) and three reviewers, one of whom, Michael S Gilmore (Reviewer #1), is a member of our Board of Reviewing Editors.

The reviewers have discussed the reviews with one another and the Reviewing Editor has drafted this decision to help you prepare a revised submission.

The study by Hullahalli et al. explores the role of the type II CRISPR-Cas systems in the evolution of multidrug resistant strains of *E. faecalis*. Although this CRISPR-Cas system harbors the RNA-guided Cas9 nuclease, which has revolutionized the genetic engineering of human cells, very little is known about how it works in native environments. A number of works addressed the function of Cas9 in bacteria, but with few exceptions, the experiments are performed in heterologous hosts, such as *E. coli* or *S. aureus*. The authors used native pheromone-responsive conjugation to introduce plasmids containing various CRISPR targets into a variety of natural and genetically altered *E. faecalis* strains, and in doing so studied how conflicts arise between CRISPR spacers and their targets – and how those conflicts are resolved. In this regard, this paper begins to fill a gap in our knowledge of these important CRISPR-Cas systems, which may lead to advances in their use as biotechnological tools. The possibility of reactivating the CRISPR2 locus to selectively target drug resistant *E. faecalis* strains is an attractive potential application of such technology.

The authors create the tools necessary to investigate CRISPR-Cas9 function in enterococci, which they subsequently use to show that (i) the last spacer sequence of the CRISPR array, always linked to a degenerate repeat, is not functional; (ii) that the lack of complete homology of these degenerate repeats (always the last repeat sequence of the CRISPR array) reduces homologous recombination, and thus loss of spacers within the CRISPR array; (iii) that target degradation is not immediate but gradual; and (iv), that this may be exploited to eliminate strains carrying antibiotic resistant conjugative plasmids from enterococcal populations.

The strength of the study is that it is quite thorough and systematic, and represents a significant advance in understanding of CRISPR biology in *E. faecalis.* The limitation, arguably more minor, is that it is unclear to what extent these findings may be applicable to CRISPR systems in other microbes, and is generalizable.

The following major comments should be considered:

1) The authors find that a significant portion of transconjugants appear to be escape mutants that maintain the spacer-containing plasmid targeted by the reactivated CRISPR locus. Some of these mutants have in fact lost the spacer from the CRIPPR locus. It would be of interest to know the nature of the remaining mutants (i.e., are there point mutations in the spacer target, are there second site mutations that affect CRISPR function or cas gene expression, etc.).

2) Throughout the manuscript it is hard to keep track of the different strains, genetic backgrounds and constructs. Consider adding diagrams to Figure 7, Figure 8, and 10 that show the genotypes of the recipient strains, as in Figure 3.

3) Authors note that internal degeneracy is not observed in CRISPR1 terminal repeats. They went on to prove that degeneracy in the terminal repeat of CRISPR2 causes loss of function in the terminal spacer (as inferred from the observation that conjugation frequency is equivalent, revealing lack of function of terminal spacer). But they also observed this phenomenon for the terminal spacer in the CRISPR1 locus. This seems contradictory and should be clarified, along with relating this observation to some level of selective advantage for its lack of utility.

4) Regarding the point that target degradation is not immediate but gradual, the authors present their findings as unique to enterococci (subsection “Effect of CRISPR-plasmid incompatibility on maintenance of plasmids and spacers”, last paragraph; subsection “CRISPR-mediated removal of antibiotic resistance”, first paragraph; Discussion, third and fourth paragraphs). However, it cannot be ruled out that the phenotype is due to low expression of Cas9, which would lead to inefficient cleavage. This is well documented in many genome editing studies. Since the cas9 gene was artificially introduced to activate CRISPR2 and it is not native to this strain, the results cannot be interpreted as a unique property of *E. faecalis*.

5) Regarding the proposal that "CRISPR incompatibility can be used to remove certain MDR *E. faecalis* from a population in vitro," there are several obvious limitations. One is about the importance of such technology if it can only be performed in vitro. Is that an overinterpretation? Additional clarification of how it might be applied would be helpful. As it is, it is difficult to imagine transforming strains with plasmids harboring a "conflictive" spacer, then applying antibiotic selection for the plasmid containing the targeting spacer, which will also lead to the selection of antibiotic resistant organisms, contradicting the original intent. This problem also appears at the end of the Discussion: wouldn't spreading a conjugative plasmid also promote spreading of antibiotic resistance? These internal conflicts should be resolved, since the impact of the advance is argued to derive both from the new knowledge as well as its potential for application.

6) Why do the authors believe *E. faecalis* can harbor targeting CRISPR/Cas systems and targets without dying? This is reminiscent of previous work in *P. aeruginosa* with imperfect crRNA-target sequences (see work from G. O'Toole lab). Are the crRNAs 100% complementary to the target, and secondly is the DNA damage response being triggered? The fitness cost could be a result of either slowing growth rates or increased cell death rates. Live dead staining and/or use of a DNA damage reporter would help resolve this.

---

## [Author Response]

The following major comments should be considered:

1) The authors find that a significant portion of transconjugants appear to be escape mutants that maintain the spacer-containing plasmid targeted by the reactivated CRISPR locus. Some of these mutants have in fact lost the spacer from the CRIPPR locus. It would be of interest to know the nature of the remaining mutants (i.e., are there point mutations in the spacer target, are there second site mutations that affect CRISPR function or cas gene expression, etc.).

This is an excellent comment by the reviewers. We would like to point out that our spacer deletion assays show that multiple different CRISPR alleles (including wild-type, one spacer deletion, and two spacer deletion) co-exist in some of our serial passage experiments (see Figure 4). We conclude that there are multiple different evolutionary outcomes of CRISPR conflicts with protospacers, which rationalizes a deep sequencing analysis of the overall populations.

We have undertaken this deep sequencing study, but in the *E. faecalis* T11 system, which has 21 spacers in its native CRISPR3-Cas locus. We conducted in-depth CRISPR amplicon and whole-genome sequencing analysis on T11 transconjugant populations that were forced to maintain CRISPR targets under antibiotic selection. We observe diversity in these populations at the CRISPR locus and in *cas9*. Specifically, multiple different mutant CRISPR alleles and *cas9* alleles co-occur in *E. faecalis* populations that are forced to maintain plasmids with protospacers in conflict with CRISPR-Cas. This work comprehensively addresses the question as to the multiple mechanisms by which escape can occur, using experimental evolution and population genomic approaches. This work will be submitted for publication this summer. We hope that the promise of this upcoming study, which addresses the reviewer comment in depth albeit in a different strain, will be sufficient.

We have added a sentence to the Discussion stating that a population-level genome sequencing study should be performed. We have also made several changes throughout the document to more accurately state the design and results of our spacer deletion assays.

2) Throughout the manuscript it is hard to keep track of the different strains, genetic backgrounds and constructs. Consider adding diagrams to Figure 7, Figure 8, and 10 that show the genotypes of the recipient strains, as in Figure 3.

Diagrams have been included in the figures. We hope that these diagrams assist with interpretation of the figures.

3) Authors note that internal degeneracy is not observed in CRISPR1 terminal repeats. They went on to prove that degeneracy in the terminal repeat of CRISPR2 causes loss of function in the terminal spacer (as inferred from the observation that conjugation frequency is equivalent, revealing lack of function of terminal spacer). But they also observed this phenomenon for the terminal spacer in the CRISPR1 locus. This seems contradictory and should be clarified, along with relating this observation to some level of selective advantage for its lack of utility.

CRISPR1 and CRISPR2 terminal repeats are both degenerate, but the degeneracy occurs at different nucleotide positions. This is shown in Figure 2—figure supplement 2. We apologize for the confusion on this point. We have edited the Discussion to remove the sentences that confused this point.

4) Regarding the point that target degradation is not immediate but gradual, the authors present their findings as unique to enterococci (subsection “Effect of CRISPR-plasmid incompatibility on maintenance of plasmids and spacers”, last paragraph; subsection “CRISPR-mediated removal of antibiotic resistance”, first paragraph; Discussion, third and fourth paragraphs). However, it cannot be ruled out that the phenotype is due to low expression of Cas9, which would lead to inefficient cleavage. This is well documented in many genome editing studies. Since the cas9 gene was artificially introduced to activate CRISPR2 and it is not native to this strain, the results cannot be interpreted as a unique property of E. faecalis.

It is true that we have artificially introduced *cas9* into V583 to generate strain V649. However, we would like to make two clarifying points.

The first is that numerous transconjugants arise during assessments of CRISPR1-Cas anti-conjugation defense in ATCC 4200RF and OG1RF, both of which natively possess the CRISPR1-Cas system. This suggests that the relative expression of *cas9* is similar in these strains and V649, since both display large CRISPR “escape” populations. Furthermore, a previous study (PMC4894674) from our group also made similar observations for the native CRISPR3-Cas system in *E. faecalis* T11. The fact that this phenomenon is observed for two different native Type II CRISPR-Cas systems in *E. faecalis* indicates that artificially low *cas9* expression in V649 does not explain the results we have obtained with this strain.

The second is that we took care to preserve the native expression of *cas9*, to the best of our abilities, when we generated strain V649. We have been unsuccessful in identifying the transcriptional start site of CRISPR1-*cas9* in strain ATCC 4200 using primer extension. However, in unpublished studies, we have generated multiple promoter reporter constructs using the anticipated promoter region of CRISPR1-*cas9*. For insertion of *cas9* into the V583 chromosome, we included the entire intergenic region upstream of *cas9*, including the predicted transcriptional terminator that occurs distally upstream of native *cas9* and is associated with the homolog of OG1RF_10403 (this is the gene upstream of *cas9*). This region contains an active promoter, per our unpublished reporter data. In short, we believe we have preserved the native promoter of *cas9* during the construction of V649.

However, the reviewer is absolutely correct that native *cas9* expression may be insufficient to confer robust genome defense in *E. faecalis*. If this were true, it would further expand our knowledge of the biological role of native CRISPR systems in *E. faecalis.* It would suggest that *E. faecalis* has evolved a low expression of *cas9* to tolerate CRISPR targets and thereby more readily accept foreign DNA. Numerous papers that we have referenced in the text have demonstrated the inability of CRISPR targets and CRISPR systems to coexist in the same cell. In this regard, even if *cas9* expression is low, our findings still provide insight into a unique facet of *E. faecalis* CRISPR biology. This explanation has been included in the discussion. We have also mentioned that *E. faecalis* displays a phenotype similar to *P. aeruginosa* when encountering CRISPR targets, and therefore this property may not be entirely unique to enterococci.

5) Regarding the proposal that "CRISPR incompatibility can be used to remove certain MDR E. faecalis from a population in vitro," there are several obvious limitations. One is about the importance of such technology if it can only be performed in vitro. Is that an overinterpretation? Additional clarification of how it might be applied would be helpful. As it is, it is difficult to imagine transforming strains with plasmids harboring a "conflictive" spacer, then applying antibiotic selection for the plasmid containing the targeting spacer, which will also lead to the selection of antibiotic resistant organisms, contradicting the original intent. This problem also appears at the end of the Discussion: wouldn't spreading a conjugative plasmid also promote spreading of antibiotic resistance? These internal conflicts should be resolved, since the impact of the advance is argued to derive both from the new knowledge as well as its potential for application.

Pheromone-responsive plasmids (PRPs) have been well documented to be able propagate in vivoamong *E. faecalis* populations in nature and disseminate beneficial traits other than antibiotic resistance. Indeed, we are currently developing a system that deploys the entire CRISPR machinery on PRPs for in vivostudies, so that the system can be delivered to strains that may endogenously lack *cas9*. Furthermore, we plan on utilizing a PRP that encodes a bacteriocin to provide a competitive advantage, mimicking the antibiotic selection that we used in this study. This point is clarified in the Discussion. Also, we hypothesize that the spread of a naturally-occurring conjugative vector targeting antibiotic resistance genes would help prevent the spread of other conjugative vectors that harbor those antibiotic resistance genes.

Other issues to be tackled with in vivo deployment of these constructs are spatial heterogeneity and dilution rate in the intestine. We will address these points in future studies.

6) Why do the authors believe E. faecalis can harbor targeting CRISPR/Cas systems and targets without dying? This is reminiscent of previous work in P. aeruginosa with imperfect crRNA-target sequences (see work from G. O'Toole lab). Are the crRNAs 100% complementary to the target, and secondly is the DNA damage response being triggered? The fitness cost could be a result of either slowing growth rates or increased cell death rates. Live dead staining and/or use of a DNA damage reporter would help resolve this.

Yes, the crRNAs generated here are 100% complementary to the target. This is now stated in the Materials and methods.

We thank the reviewers for this insightful comment about DNA damage response. Interestingly, we do not observe induction of *recA,* a hallmark of the SOS response, in transconjugants with a conflictive CRISPR construct. As a control, we induced the DNA damage response with levofloxacin, and saw increase in *recA* transcript levels, as expected. We have also performed live/dead staining on these transconjugants as requested and saw a small but significant increase in cell death in transconjugants harboring chromosomal CRISPR targeting. These data have been included in our revised submission as Figure 7—figure supplement 1.